**Data Availability Statement:** All relevant data are within the article.

**Funding:** The authors extend their appreciation to the Deanship of Scientific Research, King Saud

# Effect of surgical face mask wearing on tear film in women with a high body mass index

**Mana A. Alanazi\*, Gamal A. El-Hiti●\*, Reem Alotaibi, Mashaaer A. Baashen, Muteb Alanazi●, Raied Fagehi●, Ali M. Masmali**

Cornea Research Chair, Department of Optometry, College of Applied Medical Sciences, King Saud University, Riyadh, Saudi Arabia

\* amana@ksu.edu.sa (MAA); gelhiti@ksu.edu.sa (GAE-H)

## Abstract

### Purpose

Due to the COVID-19 pandemic, wearing a face mask has become an essential measure to reduce the rate of virus spreading. The aim of the study was to assess the effect of wearing a surgical face mask for a short period on the tear film parameters in subjects with a high body mass index (BMI).

### Methods

Twenty-five females with a high BMI ($31.4 \pm 5.5$ kg/m$^2$) aged 18–35 years ($22.7 \pm 4.6$ years) participated in the study. In addition, a control group consisting of 25 females ($23.0 \pm 6.7$ years) with a high BMI ($29.9 \pm 4.1$ kg/m$^2$) participated in the study in which no mask was worn. The standardized patient evaluation of eye dryness (SPEED) questionnaire was completed first, followed by the phenol red thread (PRT) and tear ferning (TF) tests, before wearing the face mask. The subjects wore the face mask for 1 hour, and the measurements were performed again immediately after its removal. For the control group, the measurements were performed twice with one hour gap.

### Results

Significant (Wilcoxon test, $p < 0.05$) differences were found between the SPEED scores ($p = 0.035$) and the PRT measurement ($p = 0.042$), before and after wearing the surgical face mask. The PRT scores have improved after wearing the surgical face mask, while the dry eye symptoms detected by the SPEED questionnaire have increased. On the other hand, no significant (Wilcoxon test, $p = 0.201$) differences were found between the TF grades before and after wearing a surgical face mask. For the control group, no significant (Wilcoxon test, $p > 0.05$) differences were found between the two scores from the SPEED questionnaire and the PRT, and TF tests.

### Conclusions

Wearing a surgical face mask for a short duration leads to a change in volume and quality of tears as well as dry eye symptoms in women with a high BMI.

University for funding through the Vice Deanship of Scientific Research Chairs, Research Chair of Cornea. The funders had no role in study design, data collection and analysis, decision to publish, or preparation of the manuscript.

**Competing interests:** The authors have declared that no competing interests exist.

# Introduction

The tear film maintains the health of the eye and vision. Instability in the tear film functions can lead to symptoms of dry eye [1]. Dry eye is an ocular surface disorder that is more common in females and the elderly [2,3]. The prevalence of dry eye is increasing and affects a large proportion of the world population [4]. The most common causes of dry eye are insufficient tear secretion and excessive evaporation of tears [5]. A shortage in the lacrimal gland supply leads to ocular surface inflammation, abnormal tear volume, and an increase in tear osmolarity [6–8]. The dysfunction of the meibomian gland causes a shortage in lipid secretion that leads to a high tear evaporation rate (TER) [9]. Dry eye symptoms (e.g., itching, burning, photophobia, inflammation, and irradiation) are more apparent in individuals living in dry environments (e.g., high temperature and low humidity) [10,11]. The standardized patient evaluation of eye dryness (SPEED) questionnaire [12] can be used along with various tests such as phenol red thread (PRT) [13], tear break-up time (TBUT) [14], osmolarity [15], TER [16], tear ferning (TF) [17], and others to detect the symptoms of dry eye and status of the tear film. However, the diagnosis of dry eye is challenging and complex since no single test can give definite results. Therefore, various tests that assess different parameters should be used. Dry eye symptoms can be managed by increasing the eye's comfort through various methods [18]. The most common methods involve the use of artificial tears, anti-inflammatory drugs, ocular surgery, and avoiding harsh environments. Artificial tears are the first option to treat dry eyes. They increase the viscosity of tears and lubricant and moisten the surface of the eye [19]. Vitamin A, fatty acids, and some medications help in the reduction of dry eye inflammation [20].

Body mass index (BMI) is a simple tool to detect obesity, although increases in both body fat and lean tissue can lead to a high BMI [21]. BMI is the weight (kg) of the body of an individual divided by his or her squared height (m$^2$). A measurement of $\geq 25$ kg/m$^2$ is considered as the cut-off for a high BMI according to the World Health Organization [22]. High BMI contributes to 58% of type II diabetes, 39% of hypertension, 32% of endometrial cancer in women, 23% of ischemic stroke, 21% of ischemic heart disease, 13% of osteoarthritis, 12% of colon cancer, and 8% of postmenopausal breast cancer [22]. Moreover, a high BMI has been proven to be a risk factor for dry eye [23].

In recent years, wearing a face mask has become necessary to reduce the COVID-19 infection rate [24]. It has become a daily routine for people outdoors that has led to a huge change in our daily life habits. Therefore, the current study investigates the effect of wearing a surgical face mask for a short duration on the tear film in subjects with a high BMI.

# Materials and methods

## Study design, subjects, and ethics

This observational nonrandomized comparative study was carried out in the Clinics of the College of Applied Medical Sciences, Riyadh. The measurements took place indoors in which the temperature was fixed at 20˚C and the humidity was around 15%. The size of the sample has been calculated as 24 subjects. The probability to detect a relationship between the independent and the dependent variables at a two-sided 0.05 significance level was 80%. Twenty-five females with a high BMI (31.4 ± 5.5 kg/m$^2$) aged 18–35 years (22.7 ± 4.6 years) were recruited. In addition, a control group consisting of 25 females (23.0 ± 6.7 years) with a high BMI (29.9 ± 4.1 kg/m$^2$) participated in the study in which no mask was worn. Subjects with a history of ocular surgery, contact lens wearers, smokers, and pregnant or breastfeeding women were excluded. In addition, the exclusion criteria included subjects with thyroid gland dysfunction, a high blood cholesterol level (> 4 mmol/L), high refractive errors (more

than ± 2.00 D), vitamin A and D deficiencies, hypertension, anemia, or diabetes. The tests were performed by the same examiner at the Clinics of the College of Applied Medical Sciences, Riyadh. The SPEED questionnaire was completed first, followed by the PRT and TF tests, before wearing the surgical face mask. The Band Med three layers surgical face mask (50 pack, China) with an invisible metal strip that allows the nose to fit snugly and comfortably breathing was used. The subjects wore the face mask for 1 hour, and the measurements were performed again immediately after its removal. The masks covered the nose, and all subjects followed the protocol. For the control group, the measurements were performed twice with one hour gap for comparison. During the one-hour session, the subjects carried out no activities. The tests were carried out on the right eye of each subject, and a 5-minute gap was allowed between the tests. It is not likely that the material of the surgical face mask would induce face, skin, and eye irritation, however, such a possibility has not been tested. The current research is a preliminary study to assess the effect of wearing a face mask on the tear film in subjects with a high BMI. The study was conducted in accordance with the Declaration of Helsinki and approved by King Saud University Ethics Committee (E-21-6471). Informed written consent was obtained from all participants.

## SPEED questionnaire

The SPEED questionnaire was completed first by all subjects. It contains 12 questions in three different sections (symptoms, frequency, and severity). It is used as a validated tool for the detection of dry eye symptoms [25]. The score of the SPEED questionnaire is calculated from the frequency and severity sections [26]. The total SPEED score is obtained from the summation of the responses in the sections of frequency (from 0 to 3) and severity (from 0 to 4) divided by 28 [26]. Dry eye was defined for a SPEED questionnaire score more than 4.

## PRT test

The PRT test was performed using a cotton thread (Zone-Quick, Showa Yakuhin Kako Co, Ltd., Tokyo, Japan) containing a pH indicator. The cotton thread with a 3mm bent end was gently inserted in the lower fornix for 15 seconds. The thread changes color when it comes in contact with tears. The red-colored portion of the thread was measured, and dry eye was defined as measurements less than 10 mm [27].

## TF test

A glass capillary tube (10 μL; Merck, Schnelldorf, Germany) was used to collect a tear sample (1 μL) from the lower meniscus of the right eye. The tears collected were dried in a controlled environment (23˚C and with a humidity of 10%) for 10 minutes. A digital microscope (Olympus DP72; Tokyo, Japan) was used to observe and capture the TF patterns at a magnification of 10×. The TF patterns for each tear sample were graded based on the five-point TF grading scale (from 0 to 4), using increments of 0.1 [28–32]. Dry eye was defined for a TF grade $\geq 2$. The TF grading was done by two independent researchers in which the second one was masked. The TF scores from the two researchers were similar. The TF grades were averaged and recorded to one decimal place.

## Statistical analysis

Microsoft Excel 2016 (Microsoft Office, Microsoft Corp., Redmond, WA, USA) was used to collect the data. The Statistical Package for the Social Sciences software (IBM Software, version 22, Armonk, NY, USA) was used to analyze the data [33]. The independent samples t-test has

**Table 1. The median (IQR) for the SPEED, PRT, and TF scores in subjects with a high BMI (n = 25).**

| Parameter | Pre-wearing the mask | Post-wearing the mask | p value |
|---|---|---|---|
| SPEED* | 5.0 (7.5) | 3.1 (7.5) | 0.035 |
| PRT (mm)* | 23.0 (12.5) | 27.6 (9.5) | 0.042 |
| TF | 1.7 (0.8) | 1.6 (0.9) | 0.201 |

* Significant difference (Wilcoxon test).

been used to compare the measurements of tear film parameters. The data has been identified as statistically significant when $p < 0.05$. The data were abnormally distributed (Shapiro-Wilk test; $p < 0.05$). Therefore, the median (interquartile range; IQR) was used to represent the averages.

## Results

The median (IQR) for the SPEED, PRT, and TF scores for the subjects with a high BMI (pre and post-wearing the face mask) are shown in Table 1 and those for the control group (with an hour gap) are shown in Table 2.

The PRT scores have improved after wearing the surgical face mask, while the dry eye symptoms detected by the SPEED questionnaire have increased. Significant (Wilcoxon test, $p < 0.05$) differences were found between the SPEED scores (p = 0.035) and the PRT measurements (p = 0.042) in subjects with a high BMI before and after wearing a face mask. On the other hand, no significant (Wilcoxon test, p = 0.201) difference was found between the TF grades before and after wearing a surgical face mask. For the control group, no significant (Wilcoxon test, $p > 0.05$) differences were found between the two scores from the SPEED questionnaire and the PRT, and TF tests.

The SPEED scores recorded after wearing the mask had increased in 6 subjects (24%), decreased in 10 (40%), and were unchanged in 9 cases (36%). For the PRT, the score increased in 13 subjects (52%), decreased in 8 subjects (32%), and was unchanged in four cases after wearing the mask. For the TF test, the grade after wearing the mask increased in the majority of subjects (n = 14; 56%) and decreased in 11 cases (44%). The TF test indicated that the quality of tears was reduced in the majority of subjects in the study group as a result of wearing the mask. Clearly, the correlations between the SPEED, TF, and PRT scores are weak since each one of them assesses a different parameter. For the control group, no change in the quality or volume of tears was observed.

The side-by-side boxplots for the SPEED, PRT, and TF scores in subjects with a high BMI (before and after wearing a face mask) are shown in Figs 1–3, respectively. Representative TF patterns of the tears collected from four subjects before (a, c, e, and g, respectively) and after (b, d, f, and h, respectively) wearing a surgical face mask are shown in Fig 4.

**Table 2. The median (IQR) for the SPEED, PRT, and TF scores in the control group (n = 25).**

| Parameter | Measurement 1 | Measurement 2 | p value* |
|---|---|---|---|
| SPEED | 5.3 (2.3) | 5.4 (2.5) | 0.268 |
| PRT (mm) | 22.5 (6.5) | 22.7 (4.6) | 0.321 |
| TF | 1.8 (0.5) | 1.8 (0.6) | 0.521 |

* Wilcoxon test.

## SPEED score

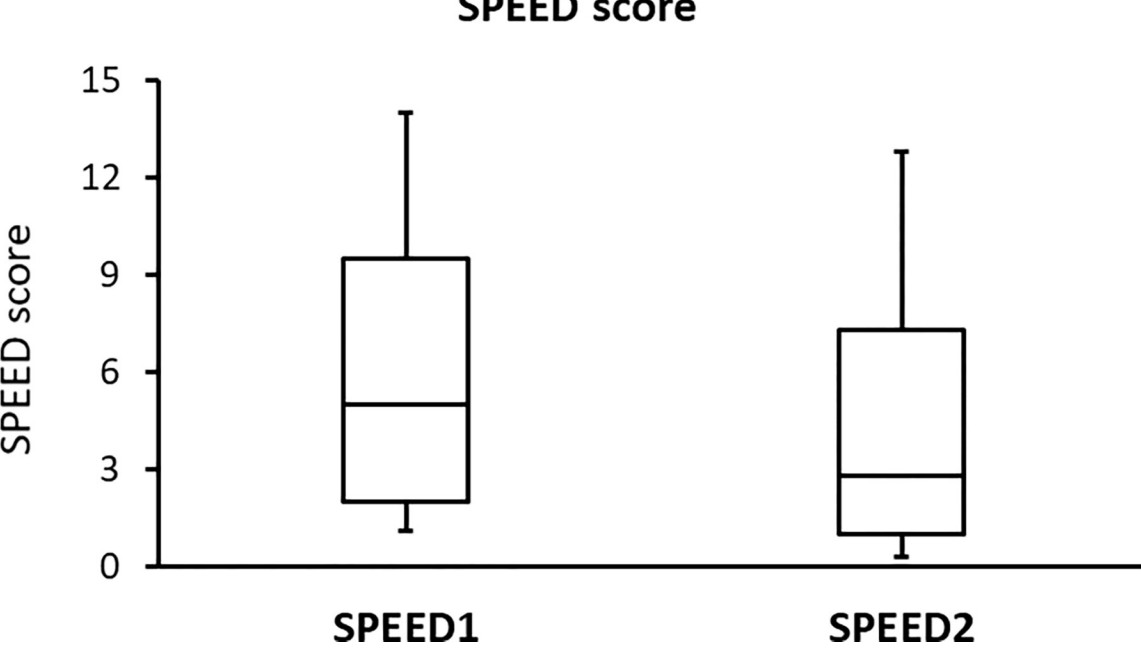

**Fig 1. Side-by-side boxplots for the SPEED scores for the subjects with a high BMI before (SPEED1) and after (SPEED2) wearing a surgical face mask.**

## PRT test

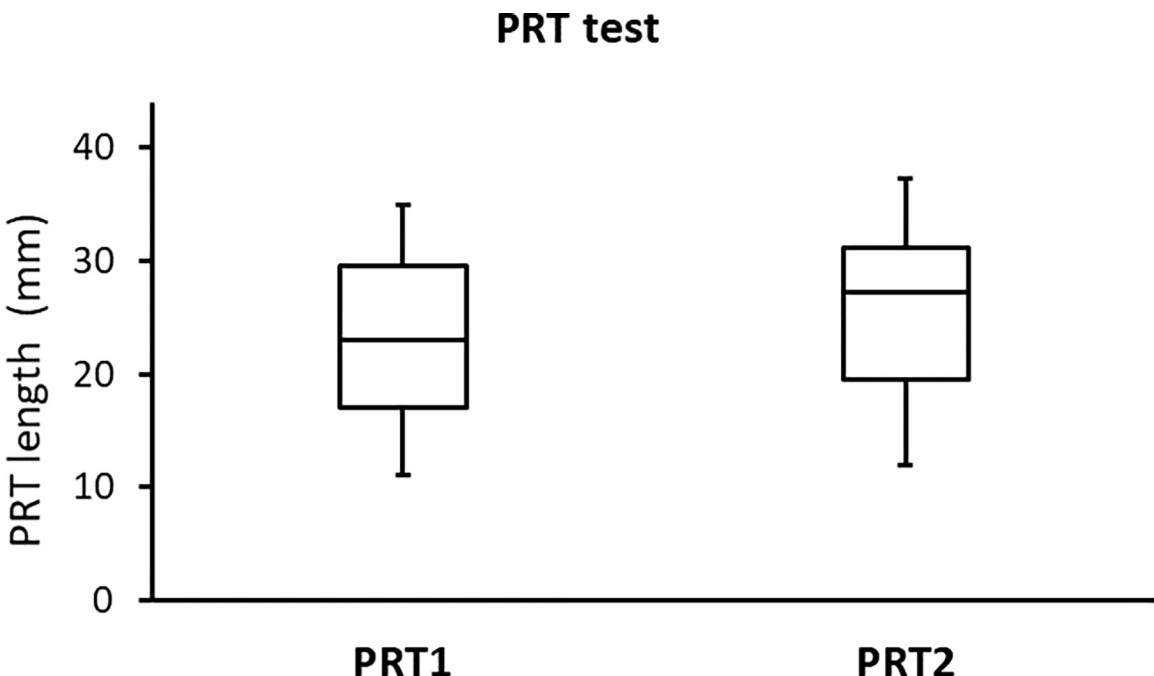

**Fig 2. Side-by-side boxplots for the PRT scores for the subjects with a high BMI before (PRT1) and after (PRT2) wearing a surgical face mask.**

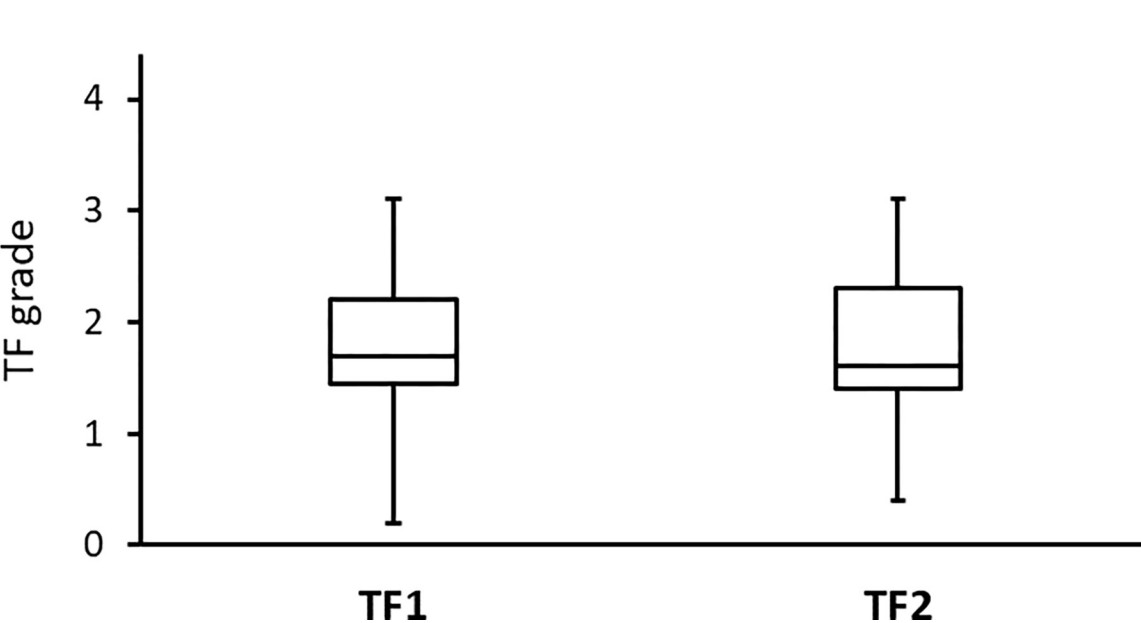

**Fig 3. Side-by-side boxplots for the TF grades for the subjects with a high BMI before (TF1) and after (TF2) wearing a surgical face mask.**

## Discussion

The current study suggests an association between wearing a medical face mask and tear film parameters. Dry eye symptoms increased significantly after wearing a face mask even for a short duration (1 hour). On the other hand, the tear volume increased significantly as a result of wearing a face mask. Dry eye has been associated with a high BMI [23,34–37]. For example, a study conducted in 20 subjects with a high BMI (31.8 (5.2) kg/m$^2$) showed a significant ($p < 0.05$) decrease in NITBUT scores and a significant increase in TF grades compared with the control group [23]. No significant ($p > 0.05$) differences were seen between the study and control groups for the ocular surface disease index (OSDI), PRT, and THM scores. In addition, a study conducted among 305 subjects using a short dry eye questionnaire suggested a medium correlation ($r = 0.34$, $p = 0.003$) between dry eye and body fat percentage [38]. However, a recent study conducted in a very large Japanese population ($n = 85,264$) that included males and females (40–74 years) suggested an inverse relationship between the prevalence of dry eye and a high BMI [39].

A recent study suggested a relationship between wearing a mask and ocular tear film instability [40]. The study was conducted among individuals wearing a mask in a medical practice in which the OSDI was used to assess dry eye symptoms. Ocular discomfort was found to be associated with wearing face masks. The severity of the symptoms was highly dependent on the type of mask used and the duration of its wearing [41]. The majority of subjects suffer from dry eye discomfort, such as burning, tearing, itching, blurred vision, and redness. The symptoms were more severe in subjects who wore regular protective medical masks compared to non-medical ones. In addition, the majority of subjects showed skin irritation, shortness of breath, physical disturbance, and headache [41]. Another cross-sectional study that was conducted with 31 subjects, mainly females ($n = 30$), showed a decrease in tear film stability after wearing face masks. The average NITBUT score was 7.8 ± 5.6 s before wearing a mask and decreased significantly ($p = 0.006$) to 6.2 ± 6.8 s afterwards [42].

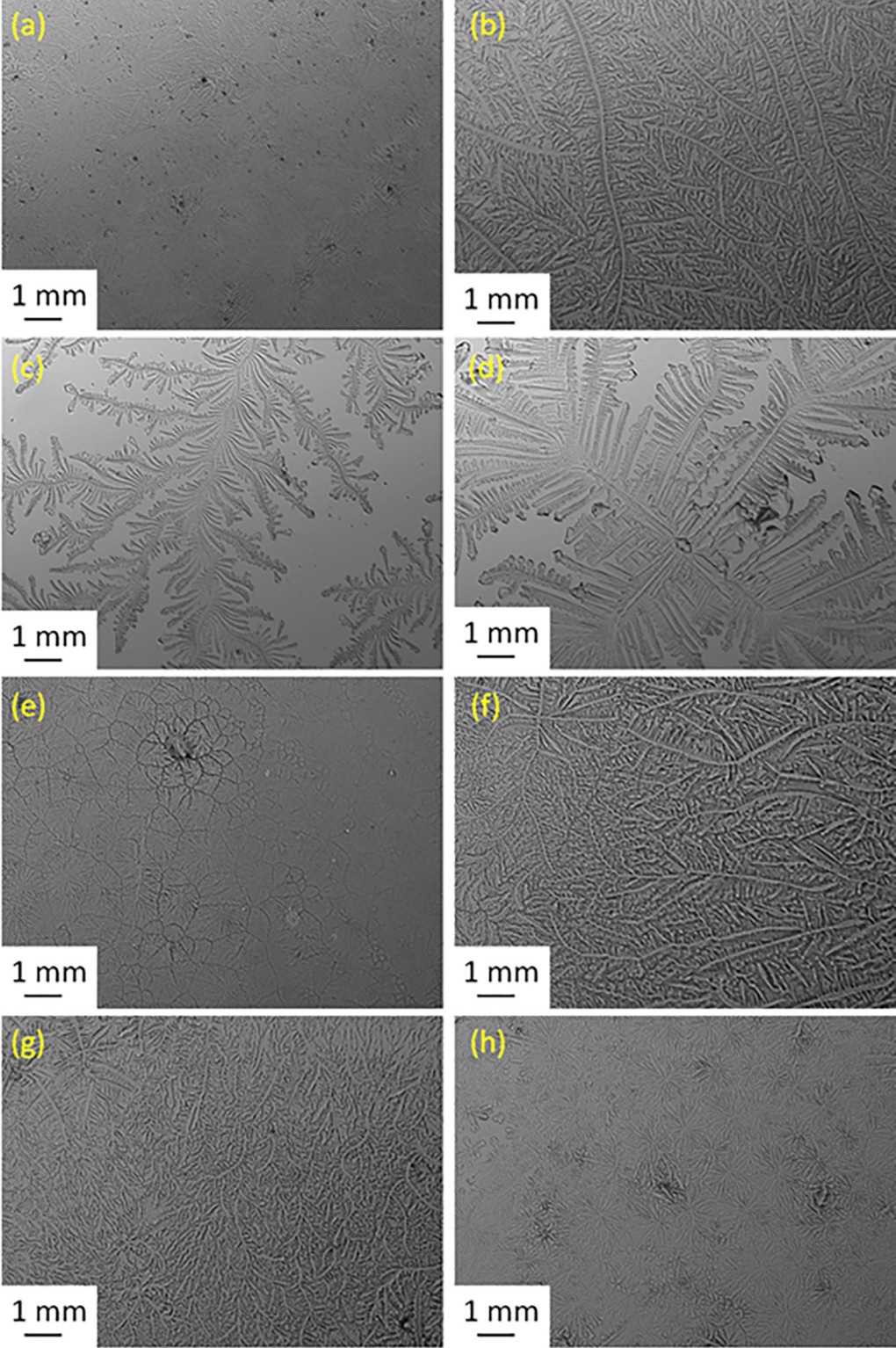

**Fig 4.** Representative TF images of the tears collected from four subjects with a high BMI before (a, c, e, and g) and after (b, d, f, and h) wearing a surgical face mask.

Very recently, it has been reported that wearing an N95 surgical face mask leads to a negative effect on the scores collected from Schirmer and TBUT tests. The tear film parameters worsen as the duration of wearing the mask increases [43]. Wearing the face mask for a long duration (e.g., more than 8 h) leads to an increase in dry eye symptoms that were detected using the OSDI a significant (p = 0.01) decrease in the TBUT [44]. Clearly, wearing a face mask for long duration has a negative effect on the tear film and daily life activities [45–47]. People infected with COVID-19 have signs of dry eye and blepharitis [48,49].

The air blowing upward from the mask into the eye might cause of dry eye symptoms. The air blowing accelerates tear evaporation, hyperosmolarity, and tear film disturbance. As a result, symptoms of ocular surface irritation and inflammation have been felt by individuals wearing masks [40–42]. The thickness of the lipid layer determines the severity of dry eye symptoms [50]. Indeed, the prevalence of dry eye was found to be high in individuals living in harsh environments due to high airflow [51,52]. Therefore, dry eye symptoms should be managed in subjects wearing face masks for a long duration. In addition, measures should be taken to prevent any ocular damage in subjects with severely dry eyes as a result of mask-wearing.

## Study limitations

The limitations of the current study include the use of a low number of subjects, the inclusion of only females from Riyadh, the short duration of mask-wearing (1 hour), and the testing of only a single type of mask (surgical). Therefore, a future study is still needed to test the effect of gender and different types of masks (e.g., surgical and fabric), and longer duration on the tear film parameters using various objective and subjective tests.

## Conclusions

Wearing a surgical face mask for a short duration leads to a change in in volume and quality of tears as well as dry eye symptoms in women with a high body mass index.

## Author Contributions

**Conceptualization:** Mana A. Alanazi, Gamal A. El-Hiti.

**Data curation:** Mana A. Alanazi, Gamal A. El-Hiti, Reem Alotaibi, Mashaaer A. Baashen, Muteb Alanazi, Raied Fagehi, Ali M. Masmali.

**Formal analysis:** Mashaaer A. Baashen, Muteb Alanazi, Raied Fagehi.

**Funding acquisition:** Gamal A. El-Hiti, Ali M. Masmali.

**Investigation:** Mana A. Alanazi, Gamal A. El-Hiti, Reem Alotaibi.

**Methodology:** Mana A. Alanazi, Gamal A. El-Hiti, Reem Alotaibi.

**Project administration:** Mana A. Alanazi.

**Resources:** Mana A. Alanazi, Gamal A. El-Hiti, Ali M. Masmali.

**Software:** Mashaaer A. Baashen, Raied Fagehi.

**Supervision:** Mana A. Alanazi, Gamal A. El-Hiti.

**Validation:** Mashaaer A. Baashen, Muteb Alanazi.

**Visualization:** Muteb Alanazi, Raied Fagehi.

**Writing – original draft:** Mana A. Alanazi, Gamal A. El-Hiti.

**Writing – review & editing:** Mana A. Alanazi, Gamal A. El-Hiti, Reem Alotaibi, Mashaaer A. Baashen, Muteb Alanazi, Raied Fagehi, Ali M. Masmali.

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
