## [Decision Letter · Decision Letter 0]

26 Sep 2022

PONE-D-22-21507Effect of surgical face mask wearing on tear film in individuals with a high body mass indexPLOS ONE

Dear Dr. El-Hiti,

Thank you for submitting your manuscript to PLOS ONE. After careful consideration, we feel that it has merit but does not fully meet PLOS ONE’s publication criteria as it currently stands. Therefore, we invite you to submit a revised version of the manuscript that addresses the points raised during the review process.

We look forward to receiving your revised manuscript.

Kind regards,

Alon Harris

Academic Editor

PLOS ONE

Journal Requirements:

   "The authors extend their appreciation to the Deanship of Scientific Research, King Saud University for funding through the Vice Deanship of Scientific Research Chairs, Research Chair of Cornea."  

    "The authors extend their appreciation to the Deanship of Scientific Research, King Saud 

University for funding through the Vice Deanship of Scientific Research Chairs, Research 

Chair of Cornea."

 "The authors extend their appreciation to the Deanship of Scientific Research, King Saud University for funding through the Vice Deanship of Scientific Research Chairs, Research Chair of Cornea."

Reviewers' comments:

Reviewer's Responses to Questions

**Comments to the Author**

1. Is the manuscript technically sound, and do the data support the conclusions?

Reviewer #1: No

Reviewer #2: Yes

2. Has the statistical analysis been performed appropriately and rigorously? 

Reviewer #1: I Don't Know

Reviewer #2: Yes

3. Have the authors made all data underlying the findings in their manuscript fully available?

Reviewer #1: Yes

Reviewer #2: Yes

4. Is the manuscript presented in an intelligible fashion and written in standard English?

Reviewer #1: Yes

Reviewer #2: Yes

5. Review Comments to the Author

Reviewer #1: The full title should be Effect of surgical face mask wearing on tear film in women with a high body mass index

Methods: high BMI (29.9 ± 4.1 years)- Should not state years but kg/m 2 instead.

What were the subjects doing in the hour between the tests? Are eyes open/closed? Same activities? Who tested the subjects? Same person? How many people analyzed the results?

Intro: They increase the viscosity of tears but result in blurry vision for a short duration [19] – This is a problematic phrasing. Furthermore, the reference does not fit. The latter involves examining two different drops on Rabbits.

A measurement of � 25

kg/m2 is considered the cut-off for a high BMI according to the WHO [22]. Abbreviation not specified.

In recent years, wearing a face mask has become necessary to reduce the COVID-

19 infection rates. Missing reference.

Methods: No specification on the way participants wore the mask – above or below the nose and who were excluded if they broke protocol.

The TF patterns for each tear sample were graded

based on the five-point TF grading scale – Was this done by a single observer or a few? This is important due to the test's nonlinear and semi-qualitative analysis.

There is no justification for the suggested number of participants in the methods section.

Results: Dry eye symptoms increased according to speed, and PRT improved. This needs further discussion. TF did not change. We do not expect the lacrimal gland to alter. How is this explained?

Discussion:

The authors suggest that parameters in the tear film changed but did not offer a hypothesis for their findings.

Conclusion –

Missing data regarding osmolarity, electrolytes, and the parameters that the writers imply changed.

In the literature review, important articles such as:

Shalaby HS, Eldesouky MEE. Effect of facemasks on the tear film during the COVID-19 pandemic. Eur J Ophthalmol. 2022 Jun 22:11206721221110010. Doi: 10.1177/11206721221110010. Epub ahead of print. PMID: 35733391; PMCID: PMC9289170.

And

Esen Baris M, Guven Yilmaz S, Palamar M. Impact of prolonged face mask wearing on tear break-up time and dry eye symptoms in health care professionals. Int Ophthalmol. 2022 Jul;42(7):2141-2144. DOI: 10.1007/s10792-022-02213-9. Epub 2022 Feb 4. PMID: 35119609; PMCID: PMC8815392

Are missing.

Reviewer #2: The authors presented an interesting study evaluating the effect of wearing a surgical face mask for a short period of time on the tear film parameters in subjects with a high body mass index. The manuscript is clear, its topic is original in content, and the conclusions are consistent with the evidence presented. The manuscript is with merit and the findings are worth reporting, but the authors should address the following comments before publication:

Introduction:

- Line 58: “tear break-up time (TBUT) [14], osmolarity [15], TER [16], tear ferning (TF) [17]” Authors need to explain the abbreviation “TER”.

Methods:

- Line 82: “Methods and methods”- please correct this repetition.

- Lines 85-86: “in a controlled environment of temperature and humidity” Authors need to explain what kind of environment was it- what temperature, what humidity, was it indoor or outdoor, were patients remaining indoor for 1 hour?

- Line 92: “high refractive errors” – the authors should expand this paragraph by providing additional details about what they mean by high refractive error?

- Line 93-94: “The medical record of each subject was checked including blood analysis.”- Authors are encouraged to specify if you checked blood work that was already done, were they looking for anything specific? Did they order any blood work if there wasn’t any results available in subjects’ charts?

Results:

- Lines 158-159: “For the control group, no significant differences were found between the two scores from the SPEED questionnaire and the PRT, and TF tests.” – Authors need to show SPEED, PRT, and TF scores for the control group.

- Line 165-166: “The TF test indicated that the quality of tears was reduced in the majority of subjects in the study group as a result of wearing the mask.” – What about control group again? Authors need to provide more details about test results in control group.

Discussion:

- Lines 192-194: “No significant differences were seen between the study and control groups for the SODI, PRT, and THM scores.” Please explain abbreviations SODI, THM as they were not used in manuscript before.

- The authors should provide discussion on the limitations of their study. For example: Is there any significance in gender differences in similar studies in the literature?

6. PLOS authors have the option to publish the peer review history of their article (what does this mean?). If published, this will include your full peer review and any attached files.

Reviewer #1: No

Reviewer #2: No

---

## [Author Response · Author response to Decision Letter 0]

28 Sep 2022

Reviewer 1

Thanks for your comments which we have answered.

The full title should be Effect of surgical face mask wearing on tear film in women with a high body mass index

The title has been modified as suggested.

Methods: high BMI (29.9 ± 4.1 years)- Should not state years but kg/m2 instead.

The error has been corrected. Now, it reads “29.9 ± 4.1 kg/m2”.

What were the subjects doing in the hour between the tests? Are eyes open/closed? Same activities? Who tested the subjects? Same person? How many people analyzed the results?

The subjects were open eyes and did no activities. The subjects were tested by the same person and the TF grading was carried out by two independent researchers. To answer the comment, the following sentence was added. “During the one-hour session, the students carried out no activities”. In addition, the following test was added at the end of the “TF test” section. “The TF grading was done by two independent researchers in which the second one was masked. The TF scores from the two researchers were similar. The TF grades were averaged and recorded to one decimal place”.

Intro: They increase the viscosity of tears but result in blurry vision for a short duration [19] – This is a problematic phrasing. Furthermore, the reference does not fit. The latter involves examining two different drops on Rabbits.

The sentence has been modified to read: “They increase the viscosity of tears and lubricant and moisten the surface of the eye”. The reference has been replaced by a more relevant one.

19. Pucker AD, Ng SM, Nichols JJ. Over the counter (OTC) artificial tear drops for dry eye syndrome. Cochrane Database Syst. Rev. 2016; 2: CD009729. https://doi.org/10.1002/14651858.CD009729.pub2 PMID: 26905373

A measurement of � 25 kg/m2 is considered the cut-off for a high BMI according to the WHO [22]. Abbreviation not specified.

The abbreviated term “WHO” has been replaced by the full name “World Health Organization”. The term has been used once so that no need to use the abbreviated name.

In recent years, wearing a face mask has become necessary to reduce the COVID-19 infection rates. Missing reference.

A new reference (shown below) was added to support the statement. The references order was amended.

24. Brooks JT, Butler JC. Effectiveness of mask wearing to control community spread of SARS-CoV-2. JAMA 2021; 325: 998–999. https://doi.org/10.1001/jama.2021.1505 PMID: 33566056

Methods: No specification on the way participants wore the mask – above or below the nose and who were excluded if they broke protocol.

The nose was covered with the masks and all subjects followed the protocol. The following sentence was added to the “Study design, subjects, and ethics” section. “The masks covered the nose and all subjects followed the protocol”.

The TF patterns for each tear sample were graded based on the five-point TF grading scale – Was this done by a single observer or a few? This is important due to the test's nonlinear and semi-qualitative analysis.

The following test was added at the end of the “TF test” section. “The TF grading was done by two independent researchers in which the second one was masked. The TF scores from the two researchers were similar. The TF grades were averaged and recorded to one decimal place”.

There is no justification for the suggested number of participants in the methods section.

The following sentence “The size of the sample has been calculated as 25 subjects” was added to the “Study design, subjects, and ethics” section.

Results: Dry eye symptoms increased according to speed, and PRT improved. This needs further discussion. TF did not change. We do not expect the lacrimal gland to alter. How is this explained?

The SPEED, PRT, and TF measure different parameters and therefore the correlation between the scores are weak. For clarification, the following sentence was added to the “Results” section. “Clearly, the correlations between the SPEED, TF, and PRT scores are weak since each one of them assesses a different parameter”.

Discussion: The authors suggest that parameters in the tear film changed but did not offer a hypothesis for their findings.

We believe that the air blowing upward from the mask into the eye might cause the changes in tear film parameter. Such a hypothesis is supported by the related results published recently. This explanation has been highlighted in the last paragraph of the “Discussion” section.

Conclusion: Missing data regarding osmolarity, electrolytes, and the parameters that the writers imply changed.

The conclusion was modified in which “… some of the tear film parameters ….” were replaced by “…. volume and quality of tears as well as dry eye symptoms …”.

In the literature review, important articles such as: Shalaby HS, Eldesouky MEE. Effect of facemasks on the tear film during the COVID-19 pandemic. Eur J Ophthalmol. 2022 Jun 22:11206721221110010. Doi: 10.1177/11206721221110010. Epub ahead of print. PMID: 35733391; PMCID: PMC9289170 and Esen Baris M, Guven Yilmaz S, Palamar M. Impact of prolonged face mask wearing on tear break-up time and dry eye symptoms in health care professionals. Int Ophthalmol. 2022 Jul;42(7):2141-2144. DOI: 10.1007/s10792-022-02213-9. Epub 2022 Feb 4. PMID: 35119609; PMCID: PMC8815392 are missing.

The two suggested references have been added and discussed in the “Discussion” section.

43. Shalaby HS, Eldesouky MEE. Effect of facemasks on the tear film during the COVID-19 pandemic. Eur J Ophthalmol 2022; ahead of print. https://doi.org/10.1177/11206721221110010. PMID: 35733391

44. Esen Baris M, Guven Yilmaz S, Palamar M. Impact of prolonged face mask wearing on tear break-up time and dry eye symptoms in health care professionals. Int Ophthalmol 2022; 42: 2141–2144. https://doi.org/10.1007/s10792-022-02213-9 PMID: 35119609

Reviewer 2

The authors presented an interesting study evaluating the effect of wearing a surgical face mask for a short period of time on the tear film parameters in subjects with a high body mass index. The manuscript is clear, its topic is original in content, and the conclusions are consistent with the evidence presented. The manuscript is with merit and the findings are worth reporting, but the authors should address the following comments before publication.

Thanks for the revision and the comments which we have answered.

Introduction:

Line 58: “tear break-up time (TBUT) [14], osmolarity [15], TER [16], tear ferning (TF) [17]” Authors need to explain the abbreviation “TER”.

The full name for the abbreviated term “TER” has been given earlier (five lines above) as “tear evaporation rate”.

Methods:

Line 82: “Methods and methods”- please correct this repetition.

The error has been corrected. Now, it reads: “Materials and methods”.

Lines 85-86: “in a controlled environment of temperature and humidity” Authors need to explain what kind of environment was it- what temperature, what humidity, was it indoor or outdoor, were patients remaining indoor for 1 hour?

To answer the comment, the text was modified to read: “The measurements took place indoors in which the temperature was fixed at 20 �C and the humidity was around 15%”. In addition, the following sentence was added. “During the one-hour session, the students carried out no activities”.

Line 92: “high refractive errors” – the authors should expand this paragraph by providing additional details about what they mean by high refractive error?

Now, the text reads: “… high refractive errors (more than � 2.00 D) …”.

Line 93-94: “The medical record of each subject was checked including blood analysis.”- Authors are encouraged to specify if you checked blood work that was already done, were they looking for anything specific? Did they order any blood work if there wasn’t any results available in subjects’ charts?

We have not carried out any blood tests ourselves but only checked the results available in the subjects’ charts. To avoid any confusion, the sentence in question was deleted.

Results:

Lines 158-159: “For the control group, no significant differences were found between the two scores from the SPEED questionnaire and the PRT, and TF tests.” – Authors need to show SPEED, PRT, and TF scores for the control group.

Table 2 was added that contains the median (IQR) for the SPEED, PRT, and TF scores in subjects of the control group (the two measurements with an hour gap).

Line 165-166: “The TF test indicated that the quality of tears was reduced in the majority of subjects in the study group as a result of wearing the mask.” – What about control group again? Authors need to provide more details about test results in control group.

The following sentence was added to answer the comment. “For the control group, no change in the quality and volume of tears was observed”.

Discussion:

Lines 192-194: “No significant differences were seen between the study and control groups for the SODI, PRT, and THM scores.” Please explain abbreviations SODI, THM as they were not used in manuscript before.

In fact, the term “SODI” should read “OSDI” which means the ocular surface disease index. The error has been corrected in which “SODI” reads now: “OSDI”. The full name was given followed by the abbreviation the first time it appears. 

The authors should provide discussion on the limitations of their study. For example: is there any significance in gender differences in similar studies in the literature?

No literature data is available about the gender effect with respect to wearing masks. The “Limitation” section has been modified for clarification.

---

## [Decision Letter · Decision Letter 1]

6 Oct 2022

PONE-D-22-21507R1Effect of surgical face mask wearing on tear film in women with a high body mass indexPLOS ONE

Dear Dr. Gaman A. El-Hiti,

Thank you for submitting your manuscript to PLOS ONE. After careful consideration, we feel that it has merit but does not fully meet PLOS ONE’s publication criteria as it currently stands. Therefore, we invite you to submit a revised version of the manuscript that addresses the points raised during the review process.

We look forward to receiving your revised manuscript.

Kind regards,

Alon Harris

Academic Editor

PLOS ONE

Reviewers' comments:

Reviewer's Responses to Questions

**Comments to the Author**

1. If the authors have adequately addressed your comments raised in a previous round of review and you feel that this manuscript is now acceptable for publication, you may indicate that here to bypass the “Comments to the Author” section, enter your conflict of interest statement in the “Confidential to Editor” section, and submit your "Accept" recommendation.

Reviewer #1: (No Response)

Reviewer #2: All comments have been addressed

2. Is the manuscript technically sound, and do the data support the conclusions?

Reviewer #1: No

Reviewer #2: Yes

3. Has the statistical analysis been performed appropriately and rigorously? 

Reviewer #1: I Don't Know

Reviewer #2: Yes

4. Have the authors made all data underlying the findings in their manuscript fully available?

Reviewer #1: No

Reviewer #2: Yes

5. Is the manuscript presented in an intelligible fashion and written in standard English?

Reviewer #1: Yes

Reviewer #2: Yes

6. Review Comments to the Author

Reviewer #1: The author did correct most of the suggestions. This article looks better.

Nevertheless, some corrections are still not self-explanatory.

Please review the attached document for further information.

Good luck.

Reviewer #2: The manuscript is clear, authors adressed all reviewers' comments, the conclusions are consistent with the evidence presented.

7. PLOS authors have the option to publish the peer review history of their article (what does this mean?). If published, this will include your full peer review and any attached files.

Reviewer #1: No

Reviewer #2: No

---

## [Author Response · Author response to Decision Letter 1]

8 Oct 2022

Reviewer 1

The author did correct most of the suggestions. This article looks better. Nevertheless, some corrections are still not self-explanatory. For example, even after the reviewer asked for a justification for the sample size, the author changed the manuscript to: "The size of the sample has been calculated as 25 subjects". There is no power analysis. There is no mention of the software used and this is not reproducible.

The text has been modified to address the comment and to justify the sample size used. Now the text reads: “The size of the sample has been calculated as 24 subjects. The probability to detect a relationship between the independent and the dependent variables at a two-sided 0.05 significance level was 80%”.

http://hedwig.mgh.harvard.edu/sample_size/js/js_associative_quant.html

Reviewer 2

The manuscript is clear, authors addressed all reviewers' comments, the conclusions are consistent with the evidence presented.

The reviewer has no further comments.

---

## [Decision Letter · Decision Letter 2]

4 Nov 2022

Effect of surgical face mask wearing on tear film in women with a high body mass index

PONE-D-22-21507R2

Dear Dr. El-Hiti,

We’re pleased to inform you that your manuscript has been judged scientifically suitable for publication and will be formally accepted for publication once it meets all outstanding technical requirements.

Kind regards,

Alon Harris

Academic Editor

PLOS ONE

Additional Editor Comments (optional):

Reviewers' comments:

Reviewer's Responses to Questions

**Comments to the Author**

1. If the authors have adequately addressed your comments raised in a previous round of review and you feel that this manuscript is now acceptable for publication, you may indicate that here to bypass the “Comments to the Author” section, enter your conflict of interest statement in the “Confidential to Editor” section, and submit your "Accept" recommendation.

Reviewer #1: All comments have been addressed

Reviewer #2: All comments have been addressed

2. Is the manuscript technically sound, and do the data support the conclusions?

Reviewer #1: Yes

Reviewer #2: Yes

3. Has the statistical analysis been performed appropriately and rigorously? 

Reviewer #1: Yes

Reviewer #2: Yes

4. Have the authors made all data underlying the findings in their manuscript fully available?

Reviewer #1: Yes

Reviewer #2: Yes

5. Is the manuscript presented in an intelligible fashion and written in standard English?

Reviewer #1: Yes

Reviewer #2: Yes

6. Review Comments to the Author

Reviewer #1: All issues were addressed.

There are no further comments regarding this manuscript from my end.

Good luck.

Reviewer #2: The manuscript is with merit, findings are worth reporting. Authors have addressed reviewers' comments, the conclusions

are consistent with the evidence presented. I do not have other comments.

7. PLOS authors have the option to publish the peer review history of their article (what does this mean?). If published, this will include your full peer review and any attached files.

Reviewer #1: No

Reviewer #2: No

---

## [Editor Report · Acceptance letter]

8 Nov 2022

PONE-D-22-21507R2 

Effect of surgical face mask wearing on tear film in women with a high body mass index 

Dear Dr. El-Hiti:

I'm pleased to inform you that your manuscript has been deemed suitable for publication in PLOS ONE. Congratulations! Your manuscript is now with our production department. 

Kind regards, 

on behalf of

Dr. Alon Harris 

Academic Editor

PLOS ONE